# Boron- and Boric Acid-Treated Titanium Implant Surfaces in Sheep Tibia: A Histologic, Histomorphometric and Mechanical Study

**DOI:** 10.3390/bioengineering9110705

**Published:** 2022-11-17

**Authors:** Nazlı Ayşeşek, Volkan Arısan, Nilüfer Bölükbaşı Balcıoğlu, Ayşe Erol, Furkan Kuruoğlu, Merva Soluk Tekkeşin, Selim Ersanlı

**Affiliations:** 1Department of Oral Implantology, Faculty of Dentistry, İstanbul University, Fatih, 34452 İstanbul, Türkiye; 2Department of Physics, Faculty of Science, İstanbul University, Fatih, 34452 İstanbul, Türkiye; 3Department of Tumor Pathology, Institute of Oncology, İstanbul University, Fatih, 34452 İstanbul, Türkiye

**Keywords:** boric acid, boron, dental implant, mechanical tests, sheep, titanium

## Abstract

The aim of this study was to compare the topographical, chemical and osseointegration characteristics of sandblasting and acid-etching (SLA) surfaces and dental implants treated by boron compounds. Titanium (Ti) disks (n = 20) were modified using boron (B) and boric acid (H_3_BO_3_) and then compared with the conventional SLA surface via surface topographic characterizations. Dental implants (3.5 mm in diameter and 8 mm in length) with the experimental surfaces (n = 96) were inserted into the tibias of six sheep, which were left to heal for 3 and 7 weeks. Histologic, histomorphometric (bone–implant contact (BIC%)) and mechanical tests (removal torque value (RTV)) were performed. The boron-coated surface (BC group) was smoother (Rz: 4.51 μm ± 0.13) than the SLA (5.86 μm ± 0.80) and the SLA-B (5.75 μm ± 0.64) groups (*p* = 0.033). After 3 weeks, the highest mean RTV was found in the SLA group (37 N/cm ± 2.87), and the difference compared with the BC group (30 N/cm ± 2.60) was statistically significant (*p* = 0.004). After 7 weeks, the mean RTV was >80 N/cm in all groups; the highest was measured in the H_3_BO_3_-treated (BS) group (89 N/cm ± 1.53) (*p* < 0.0001). No statistically significant differences were found in the BIC%s during both healing periods between the groups. H_3_BO_3_ seems to be a promising medium for dental implant osseointegration.

## 1. Introduction

Following the discovery of osseointegration, numerous types of dental implant surface treatment and modification modalities have been introduced [1,2,3], which aim to enhance the osseointegration process and decrease or ideally eliminate the time for which the patient is asked to live without masticatory function in the corresponding edentulous region [4]. In general, surface modification methods requiring distinctive approaches and completed in intricate steps are not preferred for industrial production due to feasibility and affordability concerns [5]. The majority of the market employs the familiar method of sand-blasting and acid-etching (SLA) [6]; the latter step is used to eliminate the undesired residues from the first [7]. Nevertheless, the so-called SLA surface implants provide a high rate of success and survival in treating all types of edentulism. Yet, some early or late-term failures remain unresolved in the clinical implantology practice [8].

Boron (B) is a bioactive trace element widespread in nature [9]. It plays critical roles in metabolic pathways, such as in calcium interaction and vitamin D and magnesium metabolism, which ultimately affects the activity of osteoblasts, osteoclasts and bone apposition [10,11]. Various forms of B compounds (i.e., H_3_BO_3_, BN, CaB, TiB and NaB) are used for treating recurrent or chronic infections [12], alveolar bone regeneration [13] and surface modifications [14]. Human exposure to B occurs via nutrition, air and various consumer products, and a maximum daily dose of 2 mg is recommended for safety [10]. Nonetheless, human oral exposure to high levels of boric acid has resulted in negligible observable toxicity, as seen in accidental poisonings of up to 88 gr, of which 90% of the cases were asymptomatic [10,11,12].

In a previous in vitro investigation, B- [15] and particularly boric acid (H_3_BO_3_)-treated Ti surfaces demonstrated improved proliferation and viability for human osteoblast cells and diminished the adherence of pathogen bacteria onto the corresponding substrates [16]. This may yield a positive clinical effect on tissues around the dental implant [17].

This study was performed to evaluate the topographical, chemical and osseointegration characteristics of Ti surfaces and dental implants treated with B compounds.

## 2. Materials and Methods

### 2.1. Power Analysis

Bone-to-implant contact (BIC%) and removal torque value (RTV) were designated as the primary outcomes, and relevant data were obtained from similar previous studies [18,19]. The power of the study was expressed as 1–β (β = probability of type II error), and an effect size of 1.978 and 1.484 was found for the BIC% and RTV, respectively. Referring to parametric pairwise comparisons, a minimum of 5.7 and 5.6 implants for each surface group were calculated to obtain a statistical power of 80% at the α = 0.05 for the BIC% and RTV, respectively. Regarding the proposed two-interval healing periods, 48 implants were decided on for the histomorphometric BIC% analysis. An additional 48 implants were employed for the quantification of the RTV in the corresponding healing intervals; therefore, finally 96 implants were decided necessary. A dedicated software was used (GPower, Düsseldorf, Germany), and the unit of the statistical calculations was the implant.

### 2.2. Surface Preparation

Twenty Ti disks (10 mm in diameter and 3 mm in height) and 96 Ti dental implants (3.5 mm in diameter and 8 mm in length) were machined by a commercial manufacturer (Noble Implant Technology, İstanbul, Turkey) (Figure 1). Four different surface modifications were applied: (a) large grit (250 μm) sandblasted with aluminum oxide (Al_2_O_3_) particles, followed by acid-etching with hydrochloric and sulfuric acid (HCl and H_2_SO_4_) (SLA surface group); (b) large grit (250 μm) sandblasted with Al_2_O_3_ particles and H_3_BO_3_ particles (1–5 μm) (Sigma-Aldrich, St. Louis, MO, USA), followed by acid-etching with hydrochloric and sulfuric acid (HCl and H_2_SO_4_) (SLA-B surface group); (c) SLA-B surfaces coated with 99.5% amorphous boron powder (B) (<1 μm) (Sigma-Aldrich, St. Louis, MO, USA) by heating at 900 °C (Lenton Furnaces, Hope, UK) for 10 h (BC surface group); and (d) SLA surfaces submerged in H_3_BO_3_ saline solution (BS surface group). The BS group implants were taken out of the H_3_BO_3_ saline solution at the stage of surgical insertion (Table 1). Samples were sterilized using 25 kgW gamma rays and ultraviolet-C light (UV-C).

### 2.3. Surface Characterization

Surface morphology was examined using a scanning electron microscope (SEM; FEI Versa 3D Dual Beam, Hillsboro, OR, USA), and the surface roughness was quantitatively evaluated using an atomic force microscope (AFM-XE 100 SPM System; Induspia 5F, Suwon, Korea; scan size 50 × 50 μm^2^). To determine the three-dimensional description of the surfaces (developed interfacial area ratio (Sdr%) and texture aspect ratio (Str); scan size 2 × 2 μm^2^ with 40×; optical zoom 10×) a confocal laser scanning microscope (CLSM; Leica TCS SPE; Leica Microsystems, Heidelberg, Germany) was used. Energy-dispersive X-ray spectroscopy (EDS; Octane Super SSD, EDAX Corp., Mahwah, NJ, USA) and X-ray photoelectron spectroscopy (XPS) analysis (K-Alphatm; Thermo Scientific, Waltham, MA, USA) were used to decide the chemical configuration of the surfaces.

### 2.4. Animal Experiment

Animal ethical committee approval was obtained from Mehmet Akif Ersoy Experimental Research and Development Center, İstanbul, Türkiye (approval no: 2018/19) with an allowance of six sheep for this experimental study. All experimental procedures were performed in compliance with the animal research guidelines of the Mehmet Akif Ersoy Experimental Research and Development Center. Six Anatolian-breed sheep (2–3 years of age, weight of 50–70 kg) were used. All animals were fasted 24 h prior to surgical procedures. The tibia was selected as the experiment site to refrain from the risks of infection and early implant loss. A block randomization list was obtained via a designated software (Randlist, Datinf, Tubingen, Germany), accounting equal distribution of surface groups to each tibia and animal for 96 implants.

All surgical procedures were performed with general anesthesia under sterile conditions. Xylazine (0.1 mg/kg (intramuscular, i.m.); Rompun, Bayer, Switzerland) and ketalar (3 mg/kg (intravenous, i.v.), Ketamin HCl, Vancouver, BC, Canada) were used for sedation. General anesthesia was accomplished using an i.v. injection of pentobarbital and maintained with 3–4% sevoflurane (Sevorane 100% Inhalation Solution, Abbvie, Istanbul, Türkiye) and 100% oxygen.

The proximal tibia region was shaved and disinfected with povidone–iodine (Batticon^®^, Adeka, Samsun, Turkey). An incision of approximately 25 cm was made, skin and fascia were incised, respectively, and muscles were dissected. The implants were placed in the proximal tibia referring to the previously established block randomization order. A periodontal probe was used to maintain 10 mm distance between the implants. A total of 16 implants (8 implants for each tibia) were inserted into the right and left tibia of each animal (Figure 2). The highest achieved insertion torque value (ITV) was measured using a surgical handpiece (Saeyang Microtech Co., Ltd., Daegu, Korea), which was manually calibrated via a torque meter.

Cover screws were inserted into the implants, and flap closure was performed by the repositioning of the muscles, the fascia and the skin, respectively. Antibiotics (Novosef 1 g, 20 mg/kg (i.m.); Zentiva, İstanbul, Türkiye) and analgesics (Melox 0.1 mg/kg, (i.m.); Nobel Drug, İstanbul, Turkey) were administered during postoperative care for 1 week. For the representation of the early and late-term healing, the six sheep were separated into two groups.

### 2.5. Fluorochrome Labeling

Fluorescence labeling was used for evaluating the dynamic bone mineralization and deposition according to the healing schedule in accordance with the guidelines proposed by van Gaalen et al. [20]. Three different-colored labels were used: (1) calcein green (GK2524 Calcein; 10 mg/kg (i.v.); Glentham Life Sciences Co., England) was administered to both healing groups on the 21st day; (2) oxytetracycline yellow (Primamycin/LA, 20 mg/kg (i.m.); Zoetis, İstanbul, Turkey) was administered to the late-healing group on the 35th day; and (3) alizarin red (GT6383—Alizarin Red S; 10 mg/kg (i.v.); Glentham Life Sciences Co., Corsham, UK) was administered to the late-healing group 3 days prior to the scarification (46th day). Fluorochrome labels were assembled according to the producers’ instructions, and the pH was established at 7.1.

### 2.6. Sacrifice and RTV Measurement

Using a high dosage of anesthesia, three animals were sacrificed after 3 weeks, and three were sacrificed after 7 weeks. The corresponding tibia region was exposed, and the implants were examined using X-rays (Faxitron OR Specimen Radiography System, Hologic, Marlborough, MA, USA). The RTV was measured immediately following the stabilization of the tibia in a dedicated bench clamp. A digital torque meter (TSD-400; Electromatic Co., Inc., Lynbrook, NY, USA) was used for precise measurement. Reversal torque was enforced, and the maximum torque (N/Cm) was registered.

### 2.7. Histologic and Histomorphometric Evaluation

After the animals were euthanized, the tibia of the sheep were removed at the specified time periods for histomorphometric analysis. Sections were prepared with nondecalcified histologic slicing system (Exact 300 CL; Exakt Apparatebau, Norderstedt, Germany). The sections were analyzed using a light microscope (Olympus BX60, Tokyo, Japan) to measure the BIC%. All measurements were made using an image analysis software (Olympus Image Analysis System; Olympus Soft Imaging Solutions GmbH, Münster, Germany). The implant surfaces were analyzed in three adjoining microscopic images. The BIC% was measured at a magnification of 40×. The calculation was performed by dividing the length of the attached bone by the length of the complete implant surface (including the whole threads, but the platform surface was excluded). All measurements were made by an independent examiner on two separate days (M.S.T), and the mean values were recorded as final. A confocal scanning laser microscope (CLSM) (Leica TCS SPE; Leica Microsystems, Heidelberg, Germany) was used for fluorescence evaluation.

### 2.8. Statistical Analysis

Data were analyzed using Statistical Packages of Social Sciences (SPSS) 25.0 software. Descriptive statistics, including mean, standard deviation (SD), median, range of quartiles and 95% confidence interval were calculated. The distribution of data was evaluated using the Shapiro–Wilk normality test. Homogeneity of variances was evaluated using Levene’s test. Analysis of variance (ANOVA) was used to compare the measurements that fit the normal distribution between the groups and provide the assumption of homogeneity of variances. The Tukey test was used for post hoc comparison. The Kruskal–Wallis test was used to compare the measurements that did not fit the normal distribution between the groups. The Mann–Whitney U test was used for pairwise comparisons of the groups, and Bonferroni correction was applied to *p* values. Group-by-time interaction was evaluated using the two-way ANOVA test. A *p* value < 0.05 indicated a statistically significant difference.

ARRIVE guidelines were referred to while preparing the manuscript.

## 3. Results

### 3.1. In Vitro Findings

In the low-magnification SEM imaging (×2000), the SLA surface revealed the typically recognized topographical features [5]. The remnants of H_3_BO_3_ particles were observed on the SLA-B surface. Nanowire-shaped dense crystallized B areas (high magnification, ×20,000) were visible on the BC surface. H_3_BO_3_ particles were visible on the surfaces in both the BS and SLA-B groups, which were distributed homogenously in the SLA-B group but were rather disordered in the BS surface group (Figure 3).

The Shapiro–Wilk test revealed non-normal distribution of the Rz values (*p* = 0.015). The differences in the Rz values between the groups were statistically significant (Kruskal–Wallis test, (chi-square (χ^2^): 10.53, *p* = 0.015)). The BC group had significantly lower Rz values (mean: 4.51 μm ± 0.13) than the SLA (mean: 5.86 μm ± 0.80) and the SLA-B (mean: 5.75 μm ± 0.64) groups (Mann–Whitney U, Bonferroni corrected *p* = 0.033 for both comparisons); (Figure 4 and Figure 5 and Table 2).

The normal distribution of the Str and Sdr% values was confirmed using the Shapiro–Wilk normality test (*p* > 0.05). The highest mean Str (0.36 ± 0.02) and Sdr% (76.32% ± 4.41) were measured in the SLA surface, and the lowest was in the BC surface (Str: 0.19 ± 0.01 and Sdr%: 62.84% ± 3.57). The differences in the Str and Sdr% values were statistically significant between the groups (ANOVA, F = 82.62, *p* < 0.0001 and F = 15.45 *p* < 0.0001 for Str and Sdr%, respectively). The SLA surface revealed significantly higher Str and Sdr% values compared with the remaining groups (Tukey test, *p* < 0.0001). The BC group demonstrated a significantly lower Str value than the remaining groups (Tukey test, *p* < 0.004) and a lower Sdr% value than the SLA group (Tukey test, *p* < 0.0001) and BS group (Tukey test, *p* < 0.04) (Figure 6 and Figure 7 and Table 3).

The B, Al, C, T, N and O elements were observed on the SLA, SLA-B and BS surfaces in the EDS analysis. Ti was not detected on the BC surface due to a dense coating of B with a nanowire-shaped morphology. The atomic percentage of the B element was 7.6% on SLA-B and 18.71% on the BC surfaces.

The O, C and Ti elements were detectable using XPS on all surfaces (except Ti on the BC surface). Cl was detected on the SLA surface only (2.6%). The B element was found only on the BC surface (14.96%). The highest amount of O was measured on the SLA-B surface (43.74%), while the lowest amount of C was measured on the SLA surface (35.55%) (Figure 8).

### 3.2. In Vivo Findings

Healing was uneventful in all animals, with no adverse reactions or inflammation or implant loss in any of them. Proper healing of all experimental sites was confirmed by the X-rays (Figure 9).

#### Mechanical Test Results

The normal distribution of the mechanical test values was confirmed using the Shapiro–Wilk normality test (*p* > 0.05). All implants achieved primary stability with an approximate mean ITV of 40 N/cm, and the differences in ITV were not statistically significant in or between any of the surface groups.

The RTV tests were successfully completed in the designated 48 implants. Two-way ANOVA revealed statistically significant differences in time (F = 4468.28, *p* < 0.0001), surface groups (F = 6.04, *p* = 0.002) and surface group x time interaction (F: 14.60, *p* < 0.0001). After 3 weeks of healing, the highest mean RTV was found in the SLA group (37,68 N/cm ± 2.87), and the differences compared with the BC group (30,15 N/cm ± 2.60) were statistically significant (Tukey test, *p* = 0.004).

After 7 weeks, the mean RTV was more than 80 N/cm in all groups, and the highest was measured in the BS group (89.46 N/cm ± 1.53). The lowest mean RTV values were measured in the SLA group (80.45 N/cm ± 2.46) with statistically significant differences with the remaining groups (Tukey test, *p* < 0.01). The BS surface demonstrated the highest mean RTV values (89.46 N/cm ± 1.53), and the differences compared with the remaining groups were statistically significant (Tukey test, *p* < 0.0001) (Figure 10 and Table 4).

### 3.3. Histology and Histomorphometry

#### 3.3.1. Light Microscopic Observations

No signs of inflammatory response, foreign body reaction or necrosis were noted in any histologic slices. The active osteoid formation was visible around all groups and all implants during both healing periods. An increased fill of new bone in between the threads was visible in all groups, especially in the 7th week. The process of osseointegration was ongoing during the early healing period (3 weeks), while it was concluding in the late-term healing (7 weeks) sections (Figure 11).

#### 3.3.2. Fluorescence Microscopic Observations

Compared with the BC groups, the intensity of the early term fluorochrome staining at the bone–implant interface and in the surrounding bone area appeared to be higher in the SLA, SLA-B and BS groups. Orange and light yellow staining indicating active late-term mineralization was especially discernable in the SLA-B, BC and BS groups.

Highest fluorescence intensity in the late-term healing period was observable in the BS surface group (Figure 12).

#### 3.3.3. Bone Histomorphometry

The normality of the BIC% measurements was confirmed using the Shapiro–Wilk normality test (*p* > 0.05). The range of BIC% was 23.13–33.0% during the early healing period and 52.49–68.58% during the late-term healing period. The change in BIC% measurements from 3 to 7 weeks was statistically significant (ANOVA, F = 74.25, *p* = 0.003).

The highest mean BIC% values in the early and late-term healing were observed in the BS (mean: 33.0% ± 5.86) and SLA-B groups (mean: 68.58% ± 11.76), respectively. However, the differences in BIC% between the groups were statistically not significant during both healing periods (Table 5).

## 4. Discussion

In this study, the surfaces with distinctive B-based modifications were analyzed and compared with SLA—the surface that has been used widely in modern implantology [6]. Detailed quantification of the deployed surface characteristics was further analyzed using in vivo mechanical and histomorphometric analyses, which finally allowed the objective comparison of the designed surfaces.

The height descriptive two-dimensional parameter Ra (or Sa for the three-dimensional counterpart) is regarded as a reference when comparing the dental implant surfaces [6]. In the present study, Ra and Sa values (0.87–1.007 μm and 1–2 μm, respectively) were similar in all groups providing that the desired moderate surface roughness was achieved. In general, all surfaces demonstrated a typical SEM geometry except the BC group, which demonstrated a distinctive manifestation. Wu et al. [21] investigated the effect of growth temperature on B morphology and reported that high growth temperature resulted in nanowire formation of B on Au-coated Si and MgO, which is in agreement with the present observation. It appeared that the presently employed high-temperature coating facilitated B nanowire formation on the Ti surface. The surface roughness value (Sz) was found to be enhanced on the SLA-B and BS surfaces as a result of the treatment with H_3_BO_3_. As for the BC surface, a dense B-film formation with nanowire-shaped morphology might have caused a significant drop in the Rz, Sdr% and Str values on the BC surface compared with others and particularly with the SLA.

The B amount detected using EDS and XPS was low most probably as a result of the poor binding energy of the B element, which complicated the detection of B. This incident was reported in previous studies that revealed peaks of low magnitude at a level around 187 eV corresponding to the bonding energy of B and B-oxides at 187–189 eV [16] and 188.5–190.2 eV [22]. Hence, the presently measured percentage of B (0.2–18.71%) was similar to that in other reports [14,22]. Based on these results, it was concluded that the chemical binding energy of B-treated surfaces yielded identical output. The absence of B on the BS group in the XPS and EDS investigations might be a consequence of the rigorous drying of the surface as a requirement of the measurement processes [16].

Removal torque forces applied in the counterclockwise direction have been used as a tool for the objective quantification of the strength of osseointegration. Furthermore, in clinical implant dentistry, resistance to rotational forces was regarded as critical, especially in implants that were early or immediately loaded [4,23]. In the present study, the highest RTV was measured in the SLA group after healing of 3 weeks. This positive feature was observed in previous studies, in which SLA-surface implants achieved a high reverse torque resistance (mean 49.84 and 115.2 N/cm after 3- and 6-week healing, respectively) in the early stages of osseointegration [18,24]. It was remarkable that the B-coated surface group (BC) characterized by a lower spatial and hybrid surface roughness demonstrated the lowest early term RTV in this study. It was proposed that a moderate Ti-surface roughness (Sa: 1–2 µm) might reinforce the micro retention and organization of the blood clot, subsequently increasing the resistance to rotational forces (RTV) in the early term healing [25,26]. Accordingly, the relatively smoother surface parameters (Rz, Str and Sdr%) measured on the BC surface might have caused a diminished mechanical lock and consequent measure of the lowest RTV.

It was apparent that the biologic effect of B on the osseointegration seemed to be initiated no earlier than the third week, thereby achieving significantly higher RTV than the SLA surface in the seventh week. This might be a result of the biologic effect of B, including increased osteoblastic activity [16], angiogenesis [27] and cell mitogenesis [11].

The positive effect of B was also noted in the BIC% measurements. Although the differences were not significant, the highest BIC% during both healing periods were measured in the BS and SLA-B groups. The fluorescence microscopic observations were also indicative of a higher bone mineral deposition in BS group for late-term healing. In a study by Witek et. al., a lower amount of BIC% (16.44% ± 7.9) was reported from the boronized-machined (a thermochemical process in which boron atoms diffuse into metals, revealing a nanocrystalline surface) implants left to heal in sheep tibia for 3 weeks [15]. It is not possible to comment about the negative outcome due to the lack of data on surface characterization (EDS, XPS and AFM) and mechanical tests (RTV or ISQ) [15]. Most probably, the B molecule was scarcely bonded onto the machined Ti surface (as observed in this and the previous studies) [16], causing an uprisal in the outcome typical of machined surfaces. A reduced amount of BIC% pertinent to machined Ti-surfaced implants is known [28], and this type of implant is not routinely used in modern implant dentistry. Therefore, they were not tested in this study. The present in vivo study was novel in exploring the BIC% and RTV of implants with a B-treated surface; therefore, the results cannot be directly justified by the previous reports.

Despite an expected increase in the BIC% in accordance with the RTV values, BIC% and RTV did not reveal any correlation in this study. A similar outcome was reported by Sennerby et al. (1992) who used screw-type implants left to heal in rabbit tibiae for 6 weeks, 3 months and 6 months, which revealed no significant associations with the recorded BIC% at relevant healing intervals. The amount of compact bone surrounding the titanium fixture was shown to be related to the resistance of the reverse torques [29]. It was proposed that the corroboration of RTV and BIC% might be related to factors other than those currently investigated, such as the site of application (tibia vs. ilium vs. jaw bone) and experimental model (sheep, dog and rabbit) [30,31].

Contrary to the positive outcomes of the B-treated surfaces for RTV in the late term, no significant differences were found in the BIC%, despite a slightly higher percentage of measured BIC% values in the B-modified groups after 7 weeks. Such incidents were merely reported by some studies [13], and this might be due to the location of the axial sections corresponding to a weak or compromised site, finally resulting in a low or discordant BIC% value. However, the lack of any inflammatory reactions or any adverse event in the histologic slices and the X-ray images of all B-treated surface groups proved the high biocompatibility of B in its present form and application [32,33,34].

It should be emphasized that the implants on the sheep tibia may not appropriately represent the outcomes in the human jaw bone due to the biologic and topographic differences [27]. Owing to the regulation of the ethical committee, the unit of the statistical calculations was the implant, but a subject-based calculation might yield different outcomes due to the clustering of multiple implants on the same sheep [35]. The sustainability of the B element on the implants following surgical insertion and healing should also be analyzed in further studies. Hence, distinct surface characteristics of the employed experimental implants—particularly BC—may have been subjected to topographical changes upon the screwing and removal stages. The popular resonance frequency analysis (RFA) measurements were not undertaken in this study as RFA seems to be highly variable, related to the platform level of the implant shoulder and cortical bone thickness [36,37].

## 5. Conclusions

Within the limits of this study, it was concluded that the presently employed surface modifications via B yielded a smoother surface than the conventional SLA, which seemed to cause a reduced resistance to reverse rotational forces (RTV) in the early term healing (3 weeks). No adverse reactions were observed in the B-treated surfaces. Nevertheless, B treatment, especially the B coating, did not provide a significant advantage over the conventional SLA in the early term healing but provided a significant resistance to rotational removal forces in the late term (7 weeks). H_3_BO_3_, as employed in the BS group, seems to be a promising medium for dental implant osseointegration and warrants further investigation to optimize the dose and the method of application onto the blasted Ti surfaces.

## Figures and Tables

**Figure 1 bioengineering-09-00705-f001:**
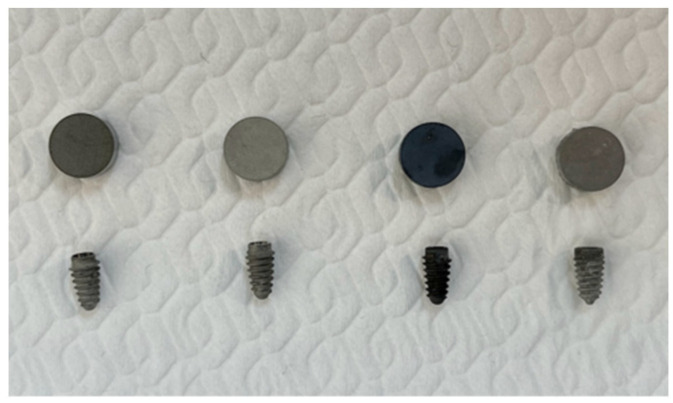
Manufactured disks and implants of the experimental surface groups.

**Figure 2 bioengineering-09-00705-f002:**
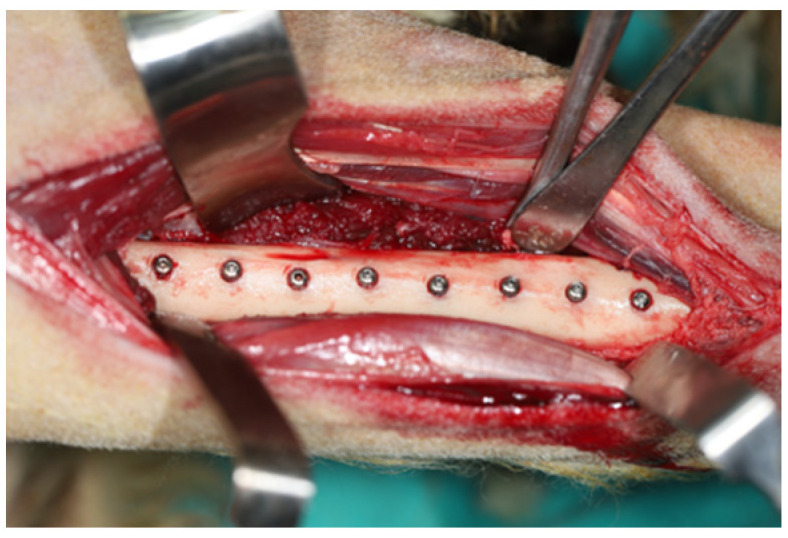
Surgical view of the sheep tibia with placed implants.

**Figure 3 bioengineering-09-00705-f003:**
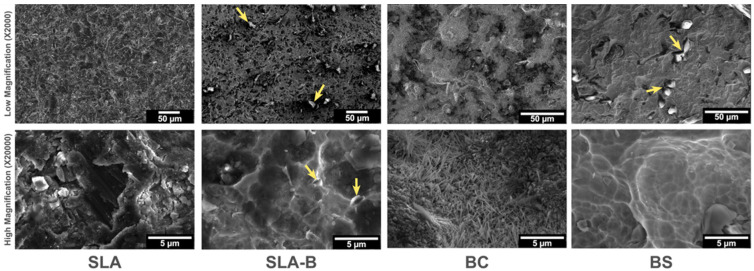
Scanning electron microscope images of the experimental surfaces. SLA surface revealed the typically known topographical features. Remnants of H_3_BO_3_ particles (arrows) were observed on the SLA-B (low magnification; ×2000). Nanowire-shaped crystallized B areas (high magnification; ×20,000) were visible on the BC surface due to the coating process at high temperatures. The BS surface was found to have nonuniformly distributed H_3_BO_3_ particles leftover from the H_3_BO_3_ solution (arrows). The region with H_3_BO_3_ particles indicated by arrows is also shown at higher magnification.

**Figure 4 bioengineering-09-00705-f004:**
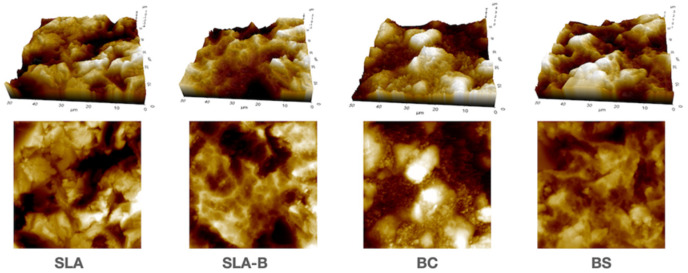
Three- and two-dimensional images of the experimental surfaces revealed using the atomic force microscope (AFM).

**Figure 5 bioengineering-09-00705-f005:**
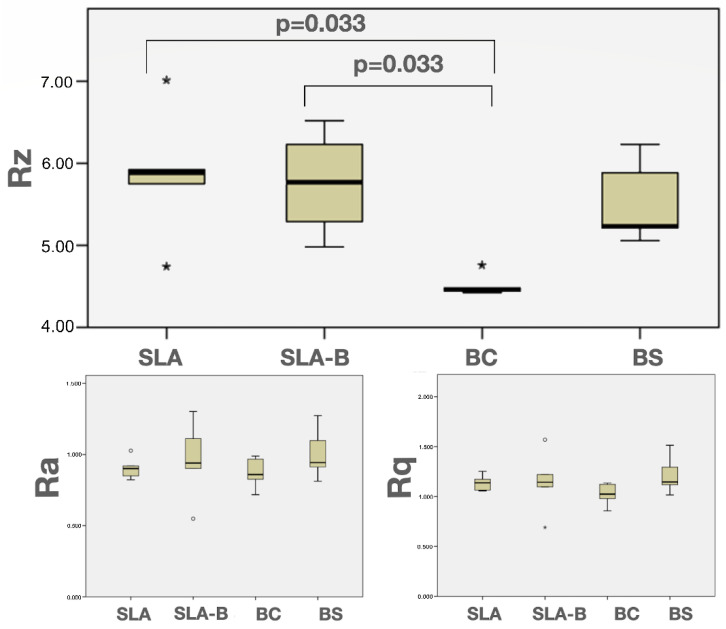
Box-and-whisker plots showing median and quartile values for the implant roughness values (Ra, Rz, and Rq). * represents outliers.

**Figure 6 bioengineering-09-00705-f006:**
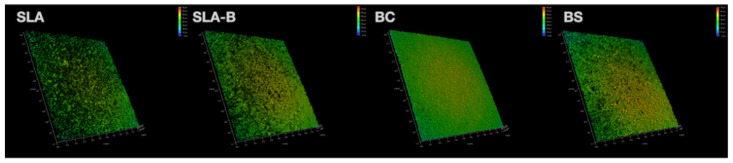
Three-dimensional images obtained by CLSM, illustrating the surface roughness and surface topography of the experimental disks.

**Figure 7 bioengineering-09-00705-f007:**
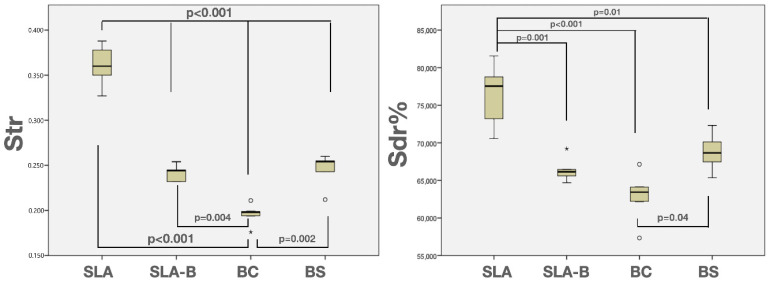
Box-and-whisker plots showing median and quartile values for the Str and Sdr% values. * represents outliers.

**Figure 8 bioengineering-09-00705-f008:**
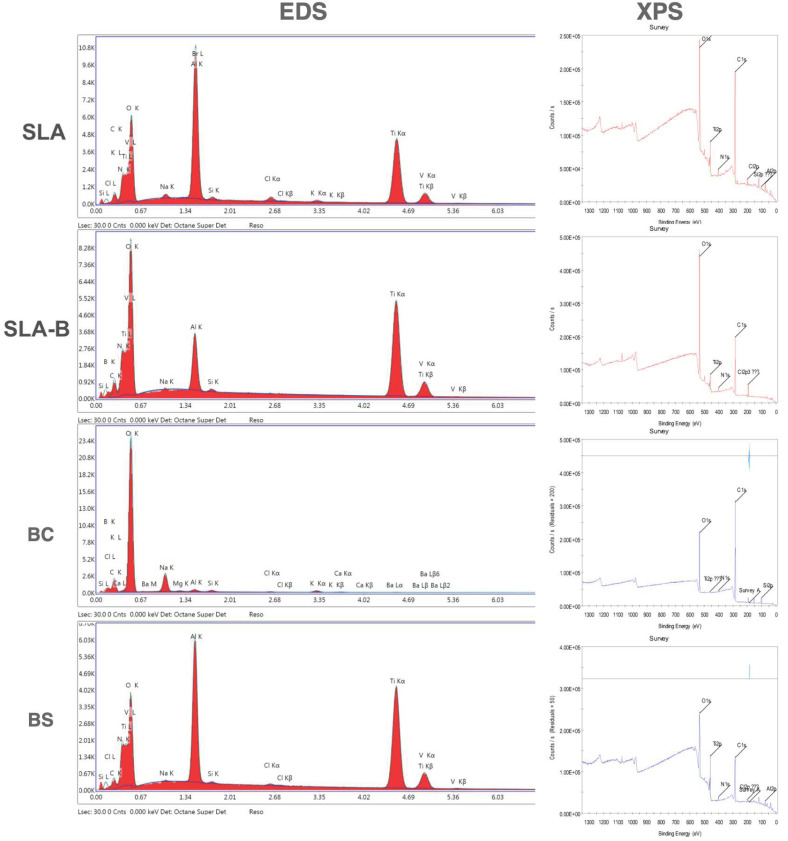
Energy-dispersive X-ray spectroscopy (EDS) and X-ray photoelectron spectroscopy (XPS) analysis of the SLA, SLA-B, BC and BS surfaces.

**Figure 9 bioengineering-09-00705-f009:**
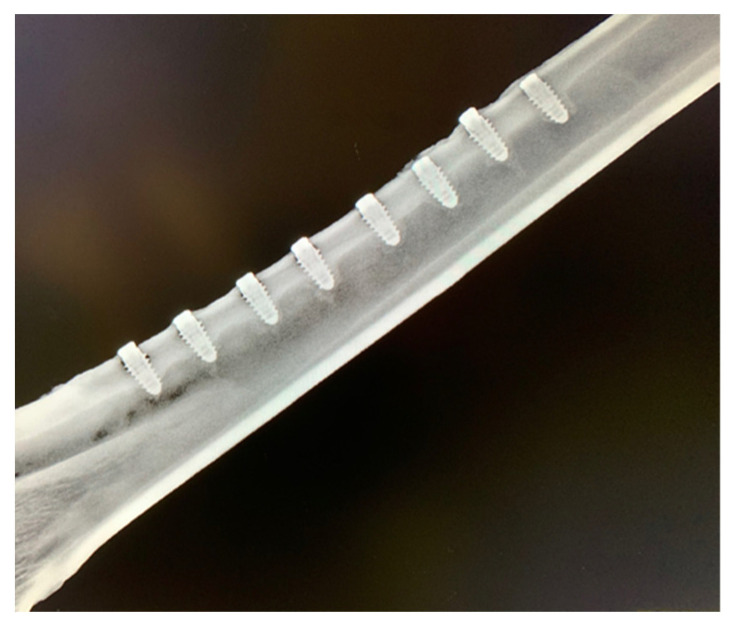
Radiographic view of the tibia and the inserted implants after 7-week healing.

**Figure 10 bioengineering-09-00705-f010:**
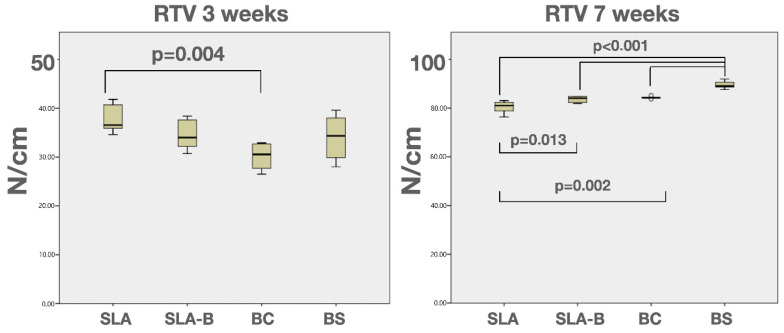
Box-and-whisker plots showing median and quartile values for the removal torque test (RTV) measurements.

**Figure 11 bioengineering-09-00705-f011:**
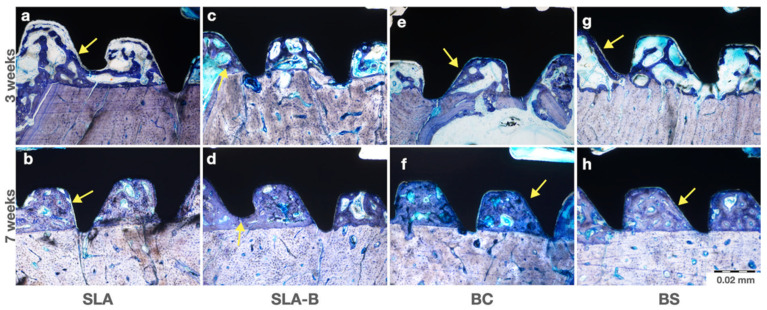
Representative histologic images of toluidine blue-stained sections in each group. Images obtained from the middle region of the implants. (**a**) The osteoid layer deposited between the old matrix and the implant surface (arrow). (**b**) The bone matrix deposition in direct contact with the surface was observed (arrow). (**c**) Active osteoid depositions were detected (arrow). (**d**) The new matrix deposition was completed; the arrow pointed to the contact line between the old and new bone matrices (arrow). (**e**) Line of osteoid deposition is also shown (arrow). (**f**) An organized bone matrix (arrow) with new matrix deposition (arrowhead) was determined. (**g**) Bone matrix deposition was evident on the implant surface (arrow). (**h**) New bone formation almost completely filled within the implant threads (arrow) (toluidine blue, original magnification ×200).

**Figure 12 bioengineering-09-00705-f012:**
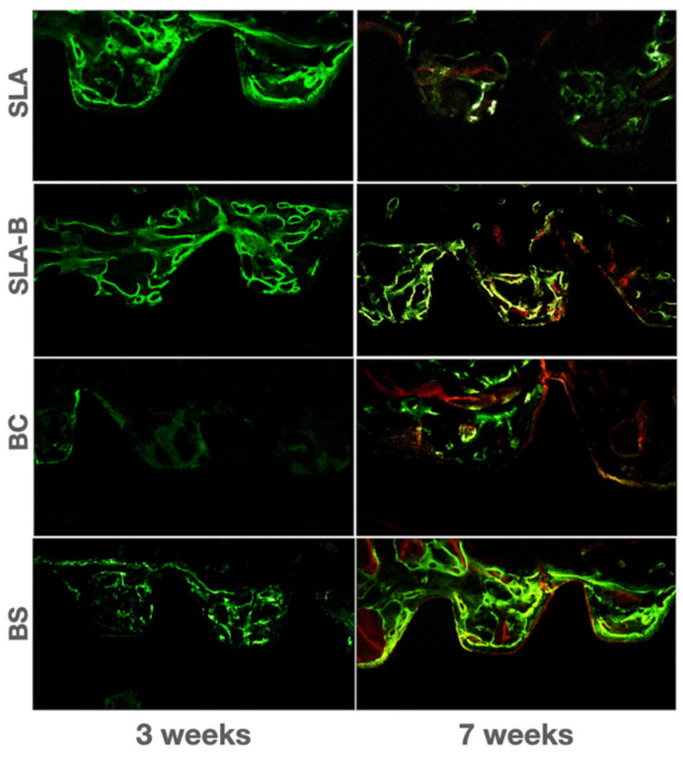
Fluorescence microscopic view of the bone deposition.

**Table 1 bioengineering-09-00705-t001:** Surface modifications of disks.

Modifications/Surface Groups	SLA	SLA-B	BC	BS
Sandblasting	Al_2_O_3_	Al_2_O_3_ + H_3_BO_3_	Al_2_O_3_ + H_3_BO_3_	Al_2_O_3_
Acid-etching	HCl + H_2_SO_4_	HCl + H_2_SO_4_	HCl + H_2_SO_4_	HCl + H_2_SO_4_
Additional treatment	-	-	B coating	H_3_BO_3_ in saline solution

**Table 2 bioengineering-09-00705-t002:** AFM measurements of the surface roughness parameters on the experimental disks.

Surface Groups	Mean (SD)	Ra (μm)	Rz * (μm)	Rq (μm)
	Median (IQR)			
	95% CI			
SLA	Mean (SD)	0.90 (0.79)	5.86 (0.80) ^a^	1.13 (0.08)
	Median (IQR)	0.90 (0.13)	5.89 (1.22)	1.13 (0.15)
	95% CI	0.80–1.00	4.85–6.86	1.03–1.23
SLA-B	Mean (SD)	0.96 (0.27)	5.75 (0.64) ^b^	1.14 (0.31)
	Median (IQR)	0.94 (0.75)	5.77 (1.25)	1.14 (0.50)
	95% CI	0.61–1.30	4.95–6.55	0.75–1.53
BC	Mean (SD)	0.87 (0.11)	4.51 (0.13) ^a,b^	1.02 (0.11)
	Median (IQR)	0.85 (0.20)	4.45 (0.18)	1.02 (0.21)
	95% CI	0.73–1.00	4.33–4.68	0.88–1.16
BS	Mean (SD)	1.007 (0.18)	5.52 (0.50)	1.21 (0.19)
	Median (IQR)	0.94 (0.32)	5.23 (0.92)	1.14 (0.33)
	95% CI	0.78–1.23	4.89–6.15	0.97–1.45

* Kruskal–Wallis test *p* = 0.015. ^a^ Mann–Whitney U test. Bonferroni corrected *p* = 0.033. ^b^ Mann–Whitney U test. Bonferroni corrected *p* = 0.033.

**Table 3 bioengineering-09-00705-t003:** Spatial (Str) and hybrid (Sdr%) surface roughness values measured using CLSM.

Surface Groups		Texture Aspect Ratio (Str) *	Developed Interfacial Area Ratio (Sdr%) *
SLA	Mean (SD)	0.36 (0.02) ^a,b,c,d^	76.32 (4.41) ^x,y,z^
	Median (IQR)	0.36 (0.04)	77.54 (8.28)
	95% CI	0.33–0.39	70.84–81.80
SLA-B	Mean (SD)	0.24 (0.009) ^a,e^	66.42 (1.69) ^x^
	Median (IQR)	0.24 (0.01)	66.14 (2.68)
	95% CI	0.22 –0.25	64.31–68.52
BC	Mean (SD)	0.19 (0.01) ^b,d,e,f^	62.84 (3.57) ^y,t^
	Median (IQR)	0.19 (0.03)	63.43 (5.86)
	95% CI	0.17–0.21	58.40–67.28
BS	Mean (SD)	0.24 (0.01) ^c,f^	68.78 (2.64) ^z,t^
	Median (IQR)	0.25 (0.03)	68.66 (4.81)
	95% CI	0.22–0.26	65.50–72.06

* ANOVA, *p* < 0.0001. ^a,b,c^ Tukey test, Pa, b, c < 0.0001. ^d,e,f^ Tukey test, Pd < 0.0001, Pe = 0.004, Pf = 0.002. ^x,y,z^ Tukey test, Px = 0.001, Py < 0.0001, Pz = 0.01. ^t^ Tukey test, Pt = 0.04.

**Table 4 bioengineering-09-00705-t004:** Descriptive statistics of the measured insertion and removal torque values (ITV and RTVs).

		SLA	SLA-B	BC	BS
ITV	Mean (SD)	42.37 (13.39)	39.05 (8.45)	42.44 (10.48)	41.63 (11.18)
	Median (IQR)	40.45 (10.15)	38.65 (7.75)	40.45 (11.01)	41.85 (17.40)
	Min–Max	27–66	18.3–69.7	29.7–68	20.9–53.2
	95% CI	36.72–48.03	35.47–42.62	38.01–46.87	36.98–46.28
RTV 3 weeks	Mean (SD)	37.68 (2.87) ^a^	34.48 (3.00)	30.15 (2.60) ^a^	34.03 (4.48)
	Median (IQR)	36.55 (5.40)	34.00 (5.47)	30.55 (5.35)	34.25 (8.98)
	Min–Max	34.6–41.8	30.7–38.4	27.7–32.9	28–39.6
	95% CI	34.66–40.70	31.32–37.63	27.42–32.87	29.32–38.73
RTV 7 weeks	Mean (SD)	80.45 (2.46) ^b,c,d,e^	83.65 (1.30) ^b,f^	84.35 (0.49) ^c,g^	89.46 (1.53) ^d,e,f,g^
	Median (IQR)	81.05 (4.30)	84.05 (2.60)	84.25 (0.60)	89.05 (2.58)
	Min–Max	76.4–83.1	81.9–84.8	83.7–85.2	87.6–91.9
	95% CI	77.86–83.03	82.28–85.01	83.83–84.86	87.85–91.08

Two-way ANOVA; time (F = 4468.28, *p* < 0.0001), surface groups (F = 6.04, *p* = 0.002) and surface group x time interaction (F: 14.60, *p* < 0.0001). ^a^ Tukey test, Pa = 0.004. ^b,c,d^ Tukey test, Pb = 0.013, Pc = 0.002, Pd < 0.0001. ^e,f,g^ Tukey test, Pe,f,g < 0.0001.

**Table 5 bioengineering-09-00705-t005:** Bone–implant contact percentages (BIC%) in the early and late-healing periods.

BIC%		SLA	SLA-B	BC	BS
3 weeks	Mean (SD)	25.82 (5.31)	23.13 (7.11)	25.95 (10.37)	33.0 (5.86)
	Median (IQR)	24.82 (5.38)	24.77 (12.61)	25.59 (18.61)	35.29 (7.74)
	Min–Max	20.19–35.94	11.35–29.56	11.28–38.97	21.76–37.38
	95% CI	20.24–31.39	15.67–30.60	15.07–36.84	26.85–39.15
7 weeks	Mean (SD)	52.49 (11.51)	68.58 (11.76)	61.34 (12.07)	64.02 (9.64)
	Median (IQR)	49.37 (20.38)	64.96 (12.23)	61.43 (16.91)	65.20 (16.42)
	Min–Max	36.84–67.79	57.18–91.26	46.91–82.30	49.32–74.54
	95% CI	40.40–64.57	56.23–80.92	48.67–74.01	53.89–74.14

Two-way ANOVA; time (F = 74.25, *p* = 0.003).

## Data Availability

The data that support the findings of this study are available from the corresponding author upon reasonable request.

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
