# Peer review of "Boron- and Boric Acid-Treated Titanium Implant Surfaces in Sheep Tibia: A Histologic, Histomorphometric and Mechanical Study"

_bioengineering, 2022, doi:10.3390/bioengineering9110705_

Round 1

Reviewer 1 Report

Please find attach my comments

Author Response

We authors thank the reviewer for the valuable comments. All raised issues has been addressed and hope that the manuscript is acceptable for publication. Please find the point-by-point responses of the authors to the reviewer's comments and relevant changes in the text.

Reviewers’ comment: #1 – The appropriate scientific correctness was not always toked into consideration. One example is:

Line 25 - Please substitute “Titanium (ti) disks (n = 20) were modified by using boron (B) and boric acid (H3BO3)” for “Titanium (Ti) disks (n = 20) were modified by using boron (B) and boric acid (H3BO3)-

Authors response: Syntax errors such as “ti / Tİ” or “H3BO3 / H3BO3” are corrected in this revision. Please note that the superscript and capitalization function of the Microsoft word may occasionally run unintentionally on pre-adjusted documents like the template of the Bioengineering and these errors might be related.

Reviewers’ comment: #2 – Although is far from my ability to correct English the entire manuscript must be revised by an English native. Otherwise it is very difficult to understand what the authors are trying to transmit. One example

Line 26 to 28 – I think that the sentence “Dental implants with the corresponding surfaces (n = 96) were inserted (3.5 mm diameter and 8 mm length) into the tibias of six sheep, which were left to heal for 3 and 7 weeks.” Is to be written: “Dental implants with the corresponding surfaces (what is corresponding surface in a dental implant??)(n = 96) were inserted into the tibias of six sheep, which were left to heal for 3 and 7 weeks.”

Authors' response: The manuscript has been revised and corrected by an English language specialist and it has been done once more according to the reviewer's request (please see attached certificate). A 200-word limit in the abstract of the Bioengineering Journal dictates the shortening of the wording in the sentences of the abstract. The “corresponding surfaces” yield to the prepared surfaces consisting of SLA and organized surfaces which have been explained and abbreviated lately in the abstract (BC and BS). This word is revised to “experimental” in this revision according to the reviewers' request.

Reviewers’ comment: #3 – Overall. What is the novelty of the study? How it contributes to the advance in the state of the art? The Introduction is too short and it does not reflect the actual state of the art: only 36 references from which only 2 from 2020; none from 2021 or 2022.

Authors response: According to the authors’ knowledge this study is the first to investigate the Boron compounds in the presently employed from on dental implant osseointegration. The introduction is improved and relevant citations are updated with citations from recent years.

Reviewers’ comment: #4 – Not valid experimental in vivo testing- Controls are missing: Ti implants without any treatment and also making holes in the tibia that must be left without any abiotic material to evaluate the normal healing process of the bone and compare the callous tissue.

Authors response: The experimental scheme is state-of-the-art and being used in numerous in vivo and in vitro analyses in similar studies [1-4]. A machined Ti implant surface is no longer used in clinical implant dentistry and the standard surface is SLA for osseointegration [5, 6]. Therefore the SLA group is serving as the control in this series of experiments (these aspects are explained in the discussion). This study aimed to investigate osteointegration -the direct contact of bone and the Ti implant- and therefore empty control defects are of no use and was not employed as was in other similar studies [2, 4].

  1. Wennerberg, A. and T. Albrektsson, On implant surfaces: a review of current knowledge and opinions. Int J Oral Maxillofac Implants, 2010. 25(1): p. 63-74.
  2. Coelho, P.G., et al., Basic research methods and current trends of dental implant surfaces. J Biomed Mater Res B Appl Biomater, 2009. 88(2): p. 579-96.
  3. Jiang, X., et al., Design of dental implants at materials level: An overview. J Biomed Mater Res A, 2020. 108(8): p. 1634-1661.
  4. Albrektsson, T. and A. Wennerberg, On osseointegration in relation to implant surfaces. Clin Implant Dent Relat Res, 2019. 21 Suppl 1: p. 4-7.
  5. Zboun, M., et al., In vitro comparison of titanium surface conditioning via boron-compounds and sand-blasting acid-etching. Surfaces and Interfaces, 2020. 21: p. 100703.
  6. Arisan, V., et al., The effect of injectable calcium phosphate cement on bone anchorage of titanium implants: an experimental feasibility study in dogs. Int J Oral Maxillofac Surg, 2010. 39(5): p. 463-8.

Reviewer 2 Report

The manuscript is reported the surface modification using boron to the Ti implant. The assessment procedure is sufficiently reported in vivo study, and is reporting enough value to be worthwhile. There are a few matters that we would like to bring to your attention and are listed below.

Introduction (discussion)

Please add any other reports on current boron properties, human toxicity, biocompatibility, and clinical applications. In current dental implant material are mainly Titanium. Boron is a material similar to ceramic, how does it compare to zirconia implants?

Material & Methods

The BS group was made by H3BO3 in saline solution. Please described the condition about solution density or dipping time.

Result

Fig 3      The SEM picture of BC was interesting. But unfortunate, the character of scale bar was not unreadable, because of resolution was low in review manuscript.

Has the author experimented study about behavior of cell migration in vitro?

Nano wire of BC is inclined to think effect to cell behavior better than SLA surface.

              Did the author verify that the structure of BC substrate was not destroyed when implant insert to tibia region? And also, removed BC implant surface was analyzed surface properties?

Result

Fig 11    the pictured area was same? Were there pictured areas of implant one is upper side, others are middle or front?

Discussion

What point in BC or BS surface modification compared to SLA or Zirconia implant? How long are coating Boron or Boric saline solution fasten SLA surface?

Author Response

We thank the reviewer for the valuable comments.

Reviewers’ comment: Introduction (discussion): Please add any other reports on current boron properties, human toxicity, biocompatibility, and clinical applications. In current dental implant material are mainly Titanium. Boron is a material similar to ceramic, how does it compare to zirconia implants?

Authors' response:  A paragraph regarding Boron, toxicity, and clinical was inserted into the introduction in this revision. There are no scientific evidence about the similarity of Boron and dental zirconia and since zirconia was not employed in this study, it was not discussed further.

Reviewers’ comment: Material & Methods: The BS group was made by H3BO3 in saline solution. Please described the condition about solution density or dipping time.

Authors' response: The implant was submerged in the H3BO3 solution and removed from the solution at the stage of surgical insertion. The required information is added to the text and can also be found in table 1.

Reviewers’ comment: Result: Fig 3 The SEM picture of BC was interesting. But unfortunate, the character of scale bar was not unreadable, because of resolution was low in review manuscript.

Has the author experimented study about behavior of cell migration in vitro? Nano wire of BC is inclined to think effect to cell behavior better than SLA surface.

Did the author verify that the structure of BC substrate was not destroyed when implant insert to tibia region? And also, removed BC implant surface was analyzed surface properties?

Authors' response: Fig 3. The resolution of the SEM images was changed to 600 dpi and the scale bars were refreshed.  

A previous study of our group investigated the in vitro cellular analyses of the boron-treated surface and this was defined by a citation in the manuscript (Zboun et al. 2020  - https://www.sciencedirect.com/science/article/abs/pii/S2468023020306957)

            Chemical and physical surface characterization following the mentioned surface treatments were performed by various surface characterization methods in the study. However, no post-removal analyses were performed. This will be included in future studies according to the valuable suggestions of the reviewer. These issues in explained in this revision as a limitation in the discussion.

Reviewers’ comment: Result: Fig 11 the pictured area was same? Were there pictured areas of implant one is upper side, others are middle or front?

Authors' response: All images were captured from the middle area of the implants. Figure legend 11 was revised with this information.

Reviewers’ comment: Discussion: What point in BC or BS surface modification compared to SLA or Zirconia implant? How long are coating Boron or Boric saline solution fasten SLA surface?

Authors' response: The observational findings regarding the BC, BS, and SLA were mentioned in the results section 3.1, Figure 3, and in the second and third paragraphs of the discussion. We believe this information is objective for the comparison of the experimental surfaces with the SLA. The sustainability of the B element following insertion and healing of the implants is a concern and will be performed in future studies. This aspect is included as a limitation in the discussion.

Reviewer 3 Report

Dear Authors, 

you made a great work!

However, some improvements are mandatory before acceptance of this manuscript. 

The paper is an original research (in vitro and animal experiment) on the Boron- and boric acid– treated titanium implant surfaces in sheep tibia: A histologic, histomorphometric and mechanical study.

The Authors made a great work in terms of methodology and the paper sounds scientific and well written.

However, some improvements are mandatory before acceptance.

The abstract is well written, complete and summary in its various aspects. The keywords are complete and appropriate.

In the introduction:

·        Even more importance it’s given to soft tissue around implants, and I think the biological seal provided by connective an epithelial tissues around implant or implant neck is actually fundamental. How does this surface act even if in contact with soft tissue? How can this solutions improve the soft tissue adaption around implants? Please improve the introduction from this point of view, considering recent research trends are going to reduce the level of inflammation in peri-implant tissues, as indicated by:
“Guarnieri R, Miccoli G, Reda R, Mazzoni A, Di Nardo D, Testarelli L. Sulcus fluid volume, IL-6, and Il-1b concentrations in periodontal and peri-implant tissues comparing machined and laser-microtextured collar/abutment surfaces during 12 weeks of healing: A split-mouth RCT. Clin Oral Implants Res. 2022 Jan;33(1):94-104. doi: 10.1111/clr.13868.”

In the materials and methods section:

·        Please explain why the removal torque value was selected as implant integration tests and not Ostell to prove implant stability? Just for have the implant removed to be studied?

·        The study is really well designed and were provided a lot of information.

Too many double spaces were found during the revision: please fix it.

Results are easy to understand and comprehensive. All the studied characteristics were reported in tables which are clear and concise.

Discussion: this section is complete and evaluates the outcome of different papers present in literature. The overall is comprehensive, concise and complete in its various aspects.

In light of the aspect underlined in the conclusion, is it possible to say, in the Authors opinion, that this surface treatment is not suggested for immediate loading use? Does the Authors suggested the use in a two stage submerged healing approach?

Conclusions are concise and clear.

Bibliography should be formatted respecting the journal’s requirements and no improper citations are evidenced.

Figures and labels are clear and easy to comprehend.

English is clear and easy to understand.

Author Response

We thank reviewer 2 for the detailed review and valuable comments.

Reviewers’ comment: In the introduction: Even more importance it’s given to soft tissue around implants, and I think the biological seal provided by connective an epithelial tissues around implant or implant neck is actually fundamental. How does this surface act even if in contact with soft tissue? How can this solutions improve the soft tissue adaption around implants? Please improve the introduction from this point of view, considering recent research trends are going to reduce the level of inflammation in peri-implant tissues, as indicated by:

“Guarnieri R, Miccoli G, Reda R, Mazzoni A, Di Nardo D, Testarelli L. Sulcus fluid volume, IL-6, and Il-1b concentrations in periodontal and peri-implant tissues comparing machined and laser-microtextured collar/abutment surfaces during 12 weeks of healing: A split-mouth RCT. Clin Oral Implants Res. 2022 Jan;33(1):94-104. doi: 10.1111/clr.13868.”

Authors' response: The introduction is revised according to the reviewers' recommendation with the relevant citation. Please note that soft tissues were not investigated in this study therefore the content was limited by the parameters employed in the material and methods.

Reviewers’ comment: In the materials and methods section:

Please explain why the removal torque value was selected as implant integration tests and not Ostell to prove implant stability? Just for have the implant removed to be studied?

Authors' response: Resonance frequency analysis exhibit values meaningful in a sequence of measurements. Single measurements could be misleading and could be directly related directly with the submergence of the implant shoulder. Therefore it was not preferred and these issues are explained in the revised discussion.

Reviewers’ comment: -Too many double spaces were found during the revision: please fix it.
Authors response: Double spaces is corrected in this revision.

Reviewers’ comment: -In light of the aspect underlined in the conclusion, is it possible to say, in the Authors opinion, that this surface treatment is not suggested for immediate loading use? Does the Authors suggested the use in a two stage submerged healing approach?

Authors' response: Immediate loading was not investigated in this study therefore commenting about immediate loading would be speculative for this study. We believe that the current form of conclusions precisely reflects the outcome derived from this experiment.

 Citations are checked and corrected for the format of the Journal

Reviewer 4 Report

The authors aimed to evaluate the topographical, chemical and osseointegration characteristics of Ti surfaces and dental implants treated with Boron (B) compounds.  The issue is important, in fact B treatment, especially B coating, did not provide a significant advantage over the conventional SLA in the early-term healing but provided a significant resistance to rotational removal forces in the late term (7 weeks). H3BO3, as employed in the BS group, seems to be a promising medium for dental implant osseointegration and warrants further investigation to optimize the dose and the method of application onto the blasted Ti surfaces. Therefore, I have no further comments against the manuscript.

Author Response

We thank the reviewer for his/her review and positive comments.